# Four Novel *Caudoviricetes* Bacteriophages Isolated from Baltic Sea Water Infect Colonizers of *Aurelia aurita*

**DOI:** 10.3390/v15071525

**Published:** 2023-07-09

**Authors:** Melissa Stante, Nancy Weiland-Bräuer, Urska Repnik, Almut Werner, Marc Bramkamp, Cynthia M. Chibani, Ruth A. Schmitz

**Affiliations:** 1Institute for General Microbiology, Christian Albrechts University, Am Botanischen Garten 1-9, D-24118 Kiel, Germany; mstante@ifam.uni-kiel.de (M.S.); nweiland@ifam.uni-kiel.de (N.W.-B.); almut-werner@web.de (A.W.); bramkamp@ifam.uni-kiel.de (M.B.); cchibani@ifam.uni-kiel.de (C.M.C.); 2Central Microscopy Facility, Christian Albrechts University, Am Botanischen Garten 1-9, D-24118 Kiel, Germany; urepnik@bio.uni-kiel.de

**Keywords:** bacteriophage, lytic cycle, Baltic Sea, *Caudoviricetes*, *Staphylococcus*, *Citrobacter*

## Abstract

The moon jellyfish *Aurelia aurita* is associated with a highly diverse microbiota changing with provenance, tissue, and life stage. While the crucial relevance of bacteria to host fitness is well known, bacteriophages have often been neglected. Here, we aimed to isolate virulent phages targeting bacteria that are part of the *A. aurita*-associated microbiota. Four phages (*Staphylococcus* phage BSwM KMM1, *Citrobacter* phages BSwM KMM2–BSwM KMM4) were isolated from the Baltic Sea water column and characterized. Phages KMM2/3/4 infected representatives of *Citrobacter*, *Shigella*, and *Escherichia* (*Enterobacteriaceae*), whereas KMM1 infected Gram-positive *Staphylococcus* spp. All phages showed an up to 99% adsorption to host cells within 5 min, short latent periods (around 30 min), large burst sizes (mean of 128 pfu/cell), and high efficiency of plating (EOP > 0.5), demonstrating decent virulence, efficiency, and infectivity. Transmission electron microscopy and viral genome analysis revealed that all phages are novel species and belong to the class of Caudoviricetes harboring a tail and linear double-stranded DNA (formerly known as *Siphovirus*-like (KMM3) and *Myovirus*-like (KMM1/2/4) bacteriophages) with genome sizes between 50 and 138 kbp. In the future, these isolates will allow manipulation of the *A. aurita*-associated microbiota and provide new insights into phage impact on the multicellular host.

## 1. Introduction

Bacteriophages, or phages for short, are viruses that infect bacteria. They are widespread in nature and are considered the Earth’s most abundant biological entities [1,2]. Phages can be classified based on their replication cycle, morphology, nucleic acid type, and host range. Particularly, their replication mode, such as virulent or temperate, is an essential characteristic [3]. Virulent phages cause lysis of the host bacterium [4], whereas temperate phages integrate their genetic material into the host bacterium’s genome and can remain dormant until conditions favor their activation into the lytic cycle [3,5,6]. In the earlier days of phage research, phages were primarily classified based on morphological characteristics. The morphological classification scheme was initially proposed by the International Committee on Taxonomy of Viruses (ICTV) which provided a framework for organizing phages into different families, genera, and species. However, with the advent of molecular biology techniques and the availability of whole-genome sequencing, the focus of phage taxonomy has shifted towards genetic and genomic information [7]. Genomic data, sequence similarities, and phylogenetic analyses assign phages to different taxonomic ranks, including families, subfamilies, genera, and species [8]. Efforts are constantly underway to improve phage taxonomy within a novel ICTV framework [9,10]. Nevertheless, morphological characteristics are still considered in the classification process and used for phage characterization.

In recent years, the interest in the role of phages in host-associated microbiomes has increased [11,12,13,14]. Researchers have recognized the significance of phages as key regulators of microbial communities and their potential impact on host health and disease [15,16]. This heightened attention reflects the growing recognition of phages as essential players in understanding the complexity and dynamics of microbiome ecosystems, particularly under the holobiont/metaorganism concept [17,18,19,20]. However, bacteriophages are still neglected [21,22,23], although the presence and activity of temperate and virulent phages in microbiomes can significantly impact the microbial community composition and function and consequently affect host health [24,25]. The temperate phage (prophage) replicates with the host genome and is transmitted to daughter cells during cell division [26]. Integrated into the host genome, they can impact the microbiome by altering the gene expression of the host bacterium, leading to changes in its physiology and metabolism [27]. Additionally, the prophage can provide additional new genes to the host bacterium and confer advantages, such as antibiotic resistance or enhanced metabolic capabilities (so-called auxiliary genes), but diseases caused directly by prophage-encoded virulence factors, such as botulism, diphtheria, and cholera, are also known to occur [28,29,30,31]. Virulent phages have the ability to reduce the population of the host bacteria in a host-associated microbiome [32], which in turn can have a significant impact on the microbiome’s structure and function, as it changes the relative abundance of different bacterial species and alters the metabolic activities of the community. Well-known examples include phages that have been shown to play an important role in the microbiomes of many invertebrates [33,34], including sponges [35], and within the squid’s light organ [36]. 

One approach to studying the role of phages in metaorganisms is metagenomic sequencing to analyze the phage populations in the microbiome [37,38,39]. Such studies can provide insights into the diversity, abundance, infection cycles (lytic or lysogenic), and activity of the phages and their interactions with the bacterial populations in the microbiome [24,40]. However, functional studies on the role of phages in microbiomes require the isolation of phages, which are, due to methodological challenges, primarily focusing on virulent phages [1,41]. Virulent phages can be isolated from various sources, including soil, water, and sewage, using a cultivation-dependent enrichment and isolation procedure. The cultivation-dependent approach includes sample collection and preparation, enrichment using selected host bacteria, isolation, and propagation [42]. Isolated novel phages can subsequently be characterized to provide a comprehensive understanding of the phage’s morphology, genetics, and behavior, which can be useful for functional studies to elucidate their role in the metaorganism and applications in phage therapy and biotechnology [43]. 

This study aimed to isolate virulent phages from Baltic Sea water infecting bacterial colonizers of our metaorganism model, the moon jellyfish *Aurelia aurita*. *A. aurita* is a Cnidarian jellyfish found in many parts of the world’s oceans, particularly common in coastal areas and estuaries [44]. *A. aurita* has a relatively short lifespan as a mature medusa, usually living only for a few months to a year [45]. During this time, *A. aurita* undergoes a complex life cycle that includes both asexual and sexual reproduction [45]. *A. aurita* is associated with a highly diverse microbiota depending on its provenance, tissue, and life stage [21]. This specific microbiota is crucial for the survival, growth, and asexual reproduction of the host [23]. A total of 132 bacterial representatives of the associated microbiota were derived from different sub-populations and life stages of *A. aurita* [22], identified by partial bacterial 16S rRNA Sanger sequencing, representing different taxonomic groups, including Proteobacteria (*Pseudomonas*, *Alteromonas*, *Pseudoalteromonas*, *Vibrio*, *Paracoccus*, *Ruegeria*, *Shewanella*, *Sulfitobacter*), Actinomycetota (*Rhodococcus*, *Brevibacterium*, *Microbacterium*, *Micrococcus*), Bacilli (*Bacillus*, *Enterococcus*, *Staphylococcus*, *Streptococcus*), and Flavobacteriia (*Chryseobacterium*, *Maribacter*, *Olleya*). Though the importance and impact of bacteria on the health and fitness of *A. aurita* have already been demonstrated [21,22,23,46], bacteriophages have not previously been considered in this context. The present study reports on the morphological, microbiological, and genomic characterization of four newly isolated virulent phages (*Staphylococcus* phage BSwM KMM1, *Citrobacter* phages BSwM KMM2, BSwS KMM3, and BSwM KMM4) infecting colonizers of *A. aurita*’s microbiota, which in the future can be used to manipulate the *A. aurita*-associated microbiota to provide new insights into phage impact on the multicellular host.

## 2. Material and Methods

### 2.1. Bacterial Strains and Growth Conditions 

The bacterial strains used in this study are listed in Table 1 with their respective culture media and growth conditions. Bacterial strains of marine origin were grown in a Marine Bouillon (MB; 10 g/L yeast extract, 10 g/L peptone (Carl Roth, Karlsruhe, Germany), 30 practical salinity units (PSU) Tropical Marine Salts, pH 7.3). Other bacterial strains were cultivated, following recommendations, in Trypticase Soy Yeast Broth (TSYB, Carl Roth, Karlsruhe, Germany), Caso-Bouillon (CASO, Carl Roth, Karlsruhe, Germany), Lysogeny Broth (LB, Carl Roth, Karlsruhe, Germany), and Nutrient Broth (NB, Carl Roth, Karlsruhe, Germany) according to the DSMZ (German Collection of Microorganisms and Cell Cultures GmbH, Braunschweig, Germany). 

### 2.2. Taxonomic Classification of Bacterial Isolates

Bacteria were enriched and isolated from *A. aurita* medusae collected in the Baltic Sea as described in the previous study by Weiland-Bräuer et al., 2020 [22]. Additional isolates previously not published were taxonomically classified in the present study. The bacterial isolates were grown, and genomic DNA was isolated from overnight cultures (5 mL) using the Wizard Genomic DNA Purification Kit (Promega GmbH, Walldorf, Germany) according to the manufacturer’s instructions. Overall, 16S rRNA genes were PCR-amplified from 50 ng of isolated genomic DNA using the bacterium-specific 16S rRNA gene primer 27F (5′-AGAGTTTGATCCTGGCTCAG-3′) and the universal primer 1492R (5′-GGTTACCTTGTTACGACTT-3′) [48] resulting in a 1.5 kb PCR fragment. The fragments were Sanger sequenced at the sequencing facility at the Institute of Clinical Molecular Biology, University of Kiel (IKMB). Sequence analysis was conducted using CodonCode Aligner v. 9.0. Sequence data of full-length 16S rRNA genes were deposited under GenBank accession numbers OQ397638-OQ397653, OQ398153-OQ398172, and PQ151711. 

### 2.3. Phage Enrichment, Isolation, and Purification

Water columns were sampled in the Baltic Sea at the Kiel fjord (54.329649, 10.149129) in March 2020 and June 2021. Samples were taken from the surface (<50 cm depth) using a sterile 20 L canister. A total of 50 mL of the seawater samples and 50 mL of MB medium were mixed with 1 mL of an overnight culture of a mixture of potential host bacterial strains (Table 1, column “Use in the study”, category “Enrichment/first screening”, a total of 55 strains) and placed on a shaking incubator (120 rpm) at 30 °C for 24 h. The mixture was centrifuged at 4000× *g* for 30 min, and the supernatant passed through a 0.22 μm pore-size polycarbonate syringe filter (Sartorius, Goettingen, Germany) to remove the residual bacterial cells. 

The spot test assay, a procedure based on the double-layer plaque technique [49], was used as an initial test to detect virulent phages by plaques on all 55 bacterial strains, separately grown in top-agar on agar plates. Briefly, 10 µL of each filtered mixture was spotted on an MB 1.5% agar plate containing a second solidified layer of 3 mL 0.6% MB top agar mixed with 100 µL of a single bacterial host strain. The plates were incubated overnight at 30 °C. Plaques generated by bacteriophage-induced bacterial lysis were detected the following day. Plaques were exclusively detected for bacterial isolates No. 8 (*Staphylococcus* sp. PQ151711), No. 7 (*Citrobacter* sp., OQ398154), and No. 6 (*C. freundii*, OQ398153). Those three bacterial isolates were used as bacterial host strains for the following assays to study phage characteristics. 

### 2.4. Phage Purification, Titration, and Propagation

Original phage plaques were used for further purification of phages. Morphologically distinct plaques were picked from the agar plate using a sterile toothpick and streaked on a freshly prepared double-agar plate with the respective host strain in the top agar. The procedure was repeated three times to ensure pure and single phages. Phage lysate preparations were conducted from top agar plates with approx. 10^5^ plaque-forming units per mL (pfu/mL). Phage-containing top agar was collected with a sterile loop and transferred into a 15 mL Falcon tube (Sarstedt, Nümbrecht, Germany). A total of 3 mL of liquid MB was added. The mixture was vortexed and centrifuged at 4000× *g* for 10 min. The supernatant was filtered through a 0.22 μm polycarbonate syringe filter (Sartorius, Goettingen, Germany) to remove bacterial cells and agar debris. The respective pfu/mL of the resulting phage lysate was determined, and the lysate was stored at 4 °C. Phage stability at 4 °C was analyzed every 2 days for 2 weeks using the double-layer agar method with no significant variance in pfu/mL.

Phage propagation was performed in liquid culture. A total of 1 mL of the respective bacterial host (overnight culture) was inoculated into 48 mL of MB in a 100 mL Erlenmeyer flask and incubated at 30 °C and 120 rpm until OD_600nm_ = 0.2–0.3 was reached. Then, 1 mL of the phage lysate (10^6^ pfu/mL) was added to the cultures, which were further incubated for 3 h at 30 °C with shaking. The culture was transferred to a 50 mL Falcon tube (Sarstedt, Germany) and centrifuged at 4000× *g* for 10 min. The supernatant was sterile-filtered using a 0.22 μm syringe filter (Sartorius, Goettingen). Phage lysates were stored at 4 °C.

### 2.5. Transmission Electron Microscopy 

In total, 50 mL of freshly prepared phage lysate (>10^8^ pfu/mL) was ultracentrifuged (Optima XE-100 ultracentrifuge, Beckman Coulter, Brea, CA, USA) at 109,800× *g* for 30 min. Phage pellets were resuspended in 1 mL of Ultra-pure water (Carl Roth, Karlsruhe, Germany) overnight at 4 °C on a 3D shaker. Subsequently, TEM grids (copper, 400 square mesh, formvar-coated) were glow-discharged for 60 s at a 0.6 mbar air pressure and a 10 mA glow current using a Safematic CCU-010 unit and then incubated with 8 μL of the phage lysate (>10^8^ pfu/mL) for 5 min. Grids were washed briefly on six drops of water, stained with 1% uranyl acetate for 10 s, blotted to remove the excess stain, and air dried. Samples were imaged with a Tecnai G2 Spirit BioTwin transmission electron microscope operated with a LaB6 filament at 80 kV, and equipped with an Eagle 4k HS CCD camera, TEM User interface (v. 4.2), and TIA software (v. 2.5) (all FEI/Thermo Fisher Scientific, Hessen, Germany). Open-source Fiji software was used to measure the head width (perpendicular to the vertical axis) and the tail length of phages.

### 2.6. One-Step Growth Curve Analysis

Bacterial host strains were grown overnight in 5 mL MB. A total of 500 µL of the overnight culture was incubated in 50 mL MB at 30 °C until turbidity at 600 nm of T_600_ = 0.1–0.2 (10^7^ cells/mL) was reached. A total of 10 mL of the bacterial culture was centrifuged at 4000× *g* for 10 min at 4 °C. The cell pellets were resuspended in 5 mL of an MB medium, and 5 mL of phage lysate (10^7^ pfu/mL) was added with a multiplicity of infection (MOI) of 1.0, expressing that one phage particle is exposed to one bacterial host cell [50,51,52]. Initial tests using MOIs of 0.01 and 0.1 did not result in a sufficient latent period and burst size detection. The phage titer was immediately determined by double-agar layer plaque assay (t_0_) after centrifuging at 4000× *g* for 10 min to remove the free phage particles before resuspending the samples in 10 mL MB. Phages were allowed to adsorb to the bacterial host cells within 5 min of incubation at 30 °C. Afterwards, the phage titer was again determined by double-agar layer plaque assay (t_5min_) for calculating the adsorption rate and constant *k* with the following formula [53]:*k* = (−1/B × t) × ln (P/P_0_). 

B, initial concentration of bacteria (cells/mL); t, time (min); P, concentration of free phage per ml; P_0_, initial concentration of phage per mL.

Further aliquots were collected in 10 min intervals over a 120 min period, and phage titers were determined. Three independent experiments were performed for each phage.

### 2.7. Efficiency of Plating and Host Range 

The host range of isolated bacteriophages was initially determined by the spot assay and verified by the double-layer agar method. A selection of 43 strains belonging to different genera (Table 1, column “Use in the study”, category “Host range”) was tested. The bacterial strains were individually grown overnight in 5 mL cultures. An aliquot of 100 μL of each culture and 3 mL of the respective culture medium containing 0.6% agar was mixed and poured onto an agar plate. After 15 min at room temperature, to allow the top agar to solidify, 10 μL of the 10-fold serially diluted phage lysate (original 10^9^ pfu/mL, diluted in MB) were spotted onto the soft agar. The plates were then incubated at the respective incubation temperature of the host strain (Table 1). Plaques were examined after 16 h of incubation. The Efficiency of Plating (EOP) was calculated by the ratio of the average pfu on a tested host to the average pfu on a corresponding reference (original) host. The variation of EOP values is represented as a heat map using Excel.

### 2.8. Viral DNA Isolation

Genomic DNA was isolated from the phage lysates using a modified phenol–chloroform–isoamyl alcohol method [54]. Briefly, 500 µL of phage lysates (10^12^ pfu/mL) were treated with 1 U/mL of DNase I and 1 U/mL of RNase A (Thermo Fisher Scientific, Hessen, Germany) in a Reaction Buffer (100 mM Tris-HCl, 25 mM MgCl_2_, 1 mM CaCl_2_) (Thermo Fisher Scientific, Hessen, Germany) and incubated for1 h at 37 °C to remove external nucleic acids. Afterwards, 0.5 M of EDTA, 40 µL of Proteinase K (20 mg/mL, Thermo Fisher Scientific, Hessen, Germany), 1 M of CaCl_2,_ and a 20% sodium dodecyl sulfate (Roth, Karlsruhe, Germany) were added before incubating at 37 °C for 2 h. Following this, samples were incubated for a further 2 h at 65 °C. Viral DNA was extracted with an equal volume (vol) of phenol–chloroform–isoamyl alcohol (25:24:1, Roth, Karlsruhe, Germany) and centrifuged at 3000× *g* for 15 min. The step was repeated, and the supernatant was transferred to phase lock tubes (Quantabio, Hilden, Germany). The aqueous phase was mixed with an equal volume of chloroform and centrifuged at 1600× *g* for 15 min. The aqueous layer was mixed with 3 M of sodium acetate and 1 vol of isopropanol to precipitate the DNA. After incubation overnight at −20 °C, the DNA was pelleted by centrifugation at 12,600× *g* for 30 min at 4 °C. The pellet was washed with a 70% ethanol, air dried, and dissolved in Ultra-pure water (Roth, Karlsruhe, Germany). DNA was stored at −20 °C before sequencing. DNA quality and quantity were analyzed using a NanoDrop1000 spectrophotometer and a Qubit double-stranded BR assay kit on a Qubit fluorometer (Thermo Fisher Scientific, Hessen, Germany).

### 2.9. Sequencing, Bioinformatic Analysis, and Annotation of Viral Genomes

Long sequencing reads were obtained using the Oxford Nanopore Technologies MinION platform (R9.4.1 flow cell). The MinION sequencing library was prepared according to the manufacturer’s guidelines using the SQK-RBK004 Rapid Barcoding Kit. MinION sequencing was performed with MinKNOW v. 22.08.4. The raw sequencing data (fast5 format) were base-called using Guppy v. 6.2.7, and finally, demultiplexing was performed using qcat v.1.1.0. Quality assessment and adapter trimming of the MinION long-read sequences of the four viral genomes was performed using LongQC v1.2.0 [55] and Filtlong v.0.2.1 (https://github.com/rrwick/Filtlong, accessed on 25 October 2022). Filtered sequence reads with an average length > 1000 bps were selected, omitting the worst 5% of reads. Reads per genome were assembled using the assembler Flye v2.9 [56], resulting in complete, single-contig genomes. The number of sequenced reads before and after filtration, the GC content, reads coverage, and N50 values are provided in Table 2. The completeness and contamination of the assembled viral genomes were assessed using CheckV [57]. 

Viral genomes were annotated using Prokka v.1.14.6 [58] and the “--kindgom Viruses” option. Functional annotation of the four isolated phages was performed using EggNOG mapper v2.1.9 [59] and the eggNOG database v5.0 [60], and the sequence searches were performed using DIAMOND [61]. Putative Auxiliary metabolic genes were additionally annotated (AMG) using DRAM-v with default databases [62], the viral mode of DRAM (Distilled and Refined Annotation of Metabolism). For this purpose, the four isolated genomes were processed through Virsorter2 v2.2.4 [63] with the “--prep-for-dramv” parameter to generate an “affi-contigs.tab” needed by DRAM-v to identify AMGs. Putative AMGs were identified based on the resulting assigned auxiliary score (AMG score ≤3) and metabolic flag (M flag, no V flag, no A flag). COG [64] and KEGG [65] annotations were derived from the EggNOG mapper and DRAM-V results. vConTACT v2.0 [66] with default settings were used to cluster the four viral genomes together with the sequences from the “ProkaryoticViralRefSeq207-Merged” to generate Viral Clusters (VCs) and determine the genus-level taxonomy of the viral genomes.

Relationships between the four isolated phages and reference genomes were analyzed using nucleic acid-based intergenomic similarities calculated with VIRIDIC (Virus Intergenomic Distance Calculator) v1.0r3.6 using default settings [67]. VIRIDIC identifies intergenomic nucleotide similarities between viruses using BLASTN pairwise comparisons and organizes viruses into clusters (genera ≥ 70% similarities and species ≥ 95% similarities). These cut-offs assign viruses into ranks following the ICTV genome identity thresholds. The reference genomes were selected based on the results of the gene-sharing network analysis where the four isolated phages clustered with viruses from the Prokaryotic Viral RefSeq Database. Genome-based phylogeny and classification of the four isolated viruses together with the same reference prokaryotic viruses were performed using the VICTOR web service (Virus Classification and Tree Building Online Resource). VICTOR is a Genome-BLAST Distance Phylogeny (GBDP) method that computes pairwise comparisons of the amino acid sequences (including 100 pseudo-bootstrap replicates) and uses them to infer a balanced minimum evolution tree with branch support via FASTME, including subtree pruning and regrafting postprocessing [68] for each of the formulas D0, D4, and D6, respectively. Trees were rooted at the midpoint [69] and visualized with ggtree [70]. The OPTSIL algorithm [71], the suggested clustering thresholds [72], and an F-value (fraction of linkages necessary for cluster fusion) of 0.5 were used to estimate taxon boundaries at the species, genus, and family levels for prokaryotic viruses [73]. The position and annotation of predicted viral genes in the phage genomes were visualized using Clinker v0.0.27 [74]. Isolated viruses were compared and visualized to the closest related species determined based on intergenomic similarity analysis. Clinker generates global alignments of amino acid sequences based on the BLOSUM62 substitution matrix. A 0.5 identity threshold was used to display the alignments. The complete phage genome sequences (assemblies) are available at NCBI under accession numbers OP902292–OP902295. Raw sequence reads were deposited on the Sequence Read Archive (SRA) under BioProject PRJNA908753 and accession numbers SRR22580853, SRR22580850, SRR22580849, and SRR22580845.

## 3. Results

Bacteriophages were isolated from the Baltic Sea water. Four novel phages infecting bacterial colonizers of the Cnidarian moon jellyfish *A. aurita* were identified and characterized. 

### 3.1. Isolation of Bacteriophages from Baltic Seawater Targeting Marine Bacteria Associated with A. aurita

Seawater samples from the Kiel fjord (Baltic Sea) were used for phage enrichments with a pool of 55 different bacteria associated with *A. aurita*. The bacteria chosen were previously described [22] and represent a diverse set of abundant species associated with *A. aurita,* possessing varying forms, colors, and colony morphologies (Table 1, [21]). One virulent phage targeting *Staphylococcus* sp. (isolate No. 8, PQ151711), one targeting *Citrobacter freundii* (isolate No. 6, OQ398153), and two phages targeting *Citrobacter* sp. (isolate No. 7, OQ398154) were isolated. Those bacteriophages were designated as *Staphylococcus* phage BSwM KMM1, *Citrobacter* phage BSwM KMM2, *Citrobacter* phage BSwS KMM3, and *Citrobacter* phage BSwM KMM4. In the following, phage designations are abbreviated to KMM1–KMM4.

### 3.2. Plaque and Virion Morphology Assign the Phages to the Class of Caudoviricetes

The identified virulent bacteriophages KMM1 (*Staphylococcus* phage), KMM2 (*Citrobacter freundii* phage), KMM3 and KMM4 (*Citrobacter* sp. phages) formed clear plaques with well-defined boundaries when infecting the respective host–bacterial strain after 16 h of incubation at 30 °C. Notably, infection with KMM3 resulted in a clear center and a turbid surrounding halo (Figure 1A and Table 3). Lysis plaques were further differentiated by halo size. Phages KMM1, KMM2, and KMM4 generated plaques with a diameter varying between 0.8 mm and 1.2 mm, while KMM3 showed plaques with a diameter of 3 mm (Figure 1A and Table 3).

Transmission Electron Microscopy (TEM) images revealed a pre-classification of all phages to the class of *Caudoviricetes*, characterized by long tails with a collar, base plates with short spikes, six long kinked tail fibers, and isometric heads (Figure 1 and Appendix A). Imaging further indicated that all phages could be assigned to the class of Caudoviricetes. KMM1, KMM2, and KMM4 showed long contractile tails typical for *Myovirus*-like phages, while KMM3 displayed a long non-contractile tail characteristic for *Siphovirus*-like phages (Figure 1, Table 3, and Appendix A). The width of the phage heads of KMM2, KMM3, and KMM4 ranged from 50 ± 1.9 nm to 58.2 ± 2.7 nm. The tail length was similar for KMM2 and KMM4, with an average length of 90.7 ± 4.2 nm, while the tail length of KMM3 was 131.7 ± 9.3 nm. Phage KMM1 was the largest of the isolated phages, with a head width of 75.8 ± 3 nm and a tail length of 176.4 ± 8.4 nm (Table 3).

### 3.3. Phages KMM1, KMM2, and KMM4 Have a Shorter Lytic Cycle Than Phage KMM3

To assess each phage’s capacity for infection, one-step growth curves were conducted with the respective host strains in an MB medium at 30 °C for 120 min in three independent biological replicates (Figure 2). The adsorption rate and constant *k* were calculated within 5 min of adsorption time, resulting in 99% of already adsorbed phage particles after 5 min leading to an adsorption constant ranging between 1.08 × 10^−7^ and 8.70 × 10^−8^ (Table 4). The calculated values for latent time and burst size are displayed in Table 3. KMM1 and KMM2 each showed an approximately 20 min latent period resulting in a burst size of 55 pfu/cell after 100 min (KMM1), while KMM2 released an average yield of 280 pfu/cell after 110 min. KMM4 infection resulted in a prolonged latent period of 30 min leading to 120 released phages (pfu/cell) after 100 min. KMM3 showed the most extended latent period with 45 min and was characterized by a burst size of 60 pfu/cell, reached after 90 min. 

### 3.4. All Isolated Phages Are Highly Specific and Effective

The KMM1 phage was initially found to infect *Staphylococcus* sp., while KMM2, KMM3, and KMM4 were shown to infect *Citrobacter* spp., which are phylogenetically classified in the *Staphylococcaceae* and *Enterobacteriaceae*, respectively. The host range of the phages was determined by spot assays on 43 strains (Table 1, column “Use in the study”, category “Host range”) belonging to the same genera, *Staphylococcus* and *Citrobacter*.

Furthermore, phages were tested against representatives of *Pseudomonadaceae, Enterobacteraceae* and *Rhodobacteraceae* of phylum Proteobacteria, *Streptococcaceae* of Bacilli, and *Chryseobacterium*, *Olleya*, and *Maribacter* of the abundant class of Flavobacteriia present in the *A. aurita*-associated microbiota. Bacterial sensitivity to a given bacteriophage was evaluated based on the occurrence of a lysis halo. Additionally, the respective phage efficiency of plating (EOP) was determined with those bacteria showing lysis in the spot tests. EOP for each host bacterium was calculated by comparing it with a score of 10^9^ pfu/mL obtained for the original host infection. As shown by the heatmap in Figure 3, KMM1 infects, in addition to the primary host, 15 additional strains of the Gram-positive family *Staphylococcaceae*, two of them even with a slightly higher EOP. The phages KMM2, KMM3, and KMM4 showed comparable and narrow host ranges within the genus *Citrobacter*. However, the observed phage titers and EOP were different as indicated by the color-coding dependent on the value (Figure 3). Phages KMM2 and KMM4 were further able to infect the *Enterobacteriaceae* bacterium *Shigella flexneri*. In contrast, phage KMM3 infected two *Escherichia coli* strains of *Enterobacteriaceae*. The phages infected none of the Flavobacteriia representatives.

### 3.5. Novel Phage Species Confirmed by Genome Sequencing Analysis 

The viral genomes of the highly effective virulent phages were sequenced using Nanopore technology. Complete phage genomes were assembled (NCBI Accession Nos. OP902292-OP902295) from Nanopore long reads of the double-stranded DNA. Table 2 summarizes the key information regarding sequencing, assembly, and annotation. Three of the four viral genomes were assigned “high-quality” (>90% completeness), while phage KMM2 was assigned as “complete” due to the presence of direct terminal repeats (DTR), which may indicate a circular genome. KMM1 has a 137 kbp genome with a GC content of 31%, while KMM2 and KMM4 showed genome sizes of approx. 87 kbp bp with an average GC content of 39%. The KMM3 genome was found to be the smallest, with 49 kbp but with the highest GC content of 43%. In total, 259 putative ORFs were predicted in the genome of phage KMM1, 137 and 138 ORFs were predicted in KMM2 and KMM4 genomes, respectively, and 92 ORFs in the KMM3 genome (Table 2). Phages were clustered into species and higher-order groups to investigate phage genomic diversity and identify closely related groups of phages. Viral Clusters (VCs) and genus-level taxonomy of the four isolated and sequenced viral genomes were generated (Appendix A). VICTOR (based on pairwise whole genome distance comparisons) was used to compare 96 previously described viral taxa with the phage genomes of KMM1–KMM4. The results indicated that the four identified phages belong to three different ranks within the class *Caudoviricetes*, as they are grouped into three different clades in the phylogenetic tree (Figure 4, Appendix A). More precisely, based on the latest ICTV classification framework, the four phages are assigned to the phylum *Uroviricota.* KMM1 assignment was resolved until the family *Herelleviridae*. KMM2 and KMM4 were classified until the genus level *Suspvirus*, belonging to the subfamily *Ounavirinae.* KMM3 belongs to the *Drexlerviridae* family, *Tempevirinae* subfamily, and subclusters into the genera *Hicfunavirus* and *Tlsvirus* due to gene homologies to both genera. Based on VICTOR and vConTACT2 analysis, the four isolated phages belong to four predicted genera and three species (Figure 4; Appendix A). VIRIDIC (Virus Intergenomic Distance Calculator) was used to determine the pairwise intergenomic similarities between the phage genomes characterized in this work compared to reference genomes. The intergenomic analysis provided evidence that the isolated viral genomes are four novel species (Appendix A, Appendix A).

Although experimental results revealed four different phages; genome analyses using VICTOR and VIRIDIC resulted in contrasting statements. Both programs confirmed that KMM1 and KMM3 are novel species. However, those analyses did not determine whether KMM2 and KMM4 were one or two species or whether they were novel. In the next step, genomes were annotated and compared to their best homologs (Figure 5). Open reading frames (ORFs) were identified, encoding basic phage-related functions, including phage DNA metabolic proteins, phage structural proteins, lysis-related proteins, and hypothetical proteins (Figure 5 and Appendix A). Genome annotations of KMM2 and KMM4 showed minor differences in their direct comparison, such as the length of genes and the presence or absence of specific genes (Figure 5B), suggesting that these are two different and novel phages, verifying phage genome analysis using VIRIDIC. 

In summary, four new phages from the Baltic Sea water column were identified and characterized that efficiently and effectively infect *Staphylococcus* and *Citrobacter* bacteria, members of the complex microbiota of the moon jellyfish *A. aurita*.

## 4. Discussion

Viruses are found in all habitats on Earth [75], but their importance is probably most evident in the ocean, where they are considered a source of diversity in genetic variation [76,77,78]. Estimates suggest that phage numbers are tenfold higher than those of bacteria in the ocean, with phage particle estimates of 10^23^, resulting in turnover rates of 10^25^ infections and lysis events per second in the ocean impacting nutrient cycling [79,80]. The relative proportions of virulent and temperate phages vary depending on various environmental factors, such as temperature, salinity, and nutrient availability [81,82]. In general, virulent phages tend to dominate in nutrient-rich environments with high microbial diversity and abundance and are thus more prevalent in surface waters, while temperate phages are more prevalent in nutrient-poor environments in deeper (oligotrophic) waters [83]. For instance, in the Baltic Sea, it has previously been demonstrated that virulent viruses are more common in surface waters, whereas lysogeny predominates in deep marine waters [77,84]. Lytic representatives of *Siphovirus*-like (52%), *Myovirus*-like (42%), and *Podovirus*-like (6%) phages of the *Caudoviricetes* class were consistent in the surface water throughout all seasons within the Baltic Sea [84,85,86,87]. Those phages can have several roles particularly in the microbiomes of marine animals and plants, e.g., to maintain a healthy microbiome and prevent the spread of diseases [24,88,89,90]. Further, phages appear responsible for several diseases that harm corals and their symbionts [91,92,93,94]. Besides, phages promote the evolution of new traits or the acquisition of beneficial genes within a microbiome by mediating horizontal gene transfer between bacteria by transduction [95,96]. Lastly, the impact mentioned above on nutrient cycling in marine ecosystems by breaking down bacterial cells and releasing nutrients into the environment can have important implications for marine organisms’ overall health and productivity [97]. Although the role of Cnidarian bacterial communities has already been intensively investigated [22,98,99,100], the impact of (virulent) phages on Cnidarian and particularly on *A. aurita*’s bacterial colonizers has yet only rarely been studied. However, different *Hydra* species have been shown to harbor a diverse host-associated virome predominated by bacteriophages [39,101,102]. Changes in environmental conditions altered the associated virome, increased viral diversity, and affected the metabolism of the metaorganism [102]. The specificity and dynamics of the virome point to a potential viral involvement in regulating microbial associations in the *Hydra* metaorganism [101]. 

In this study, four phages (*Staphylococcus* phage KMM1, *Citrobacter* phages KMM2, KMM3, and KMM4) were isolated from the Baltic Sea water column (Kiel fjord) surrounding *A. aurita* individuals by a cultivation-based approach, infecting previously isolated bacteria, *Staphylococcus* and *Citrobacter*, both present in the associated microbiota of *A. aurita* [21,22]. Phages KMM1, KMM2, and KMM4 showed a clear, roundish plaque morphology, as previously described for most *Caudoviricetes* with long contractile tails (formerly known as *Myovirus*-like phages) [103]. Phage KMM3, on the other hand, showed larger plaques with a clear center surrounded by a turbid halo, commonly referred to as a “bull’s eye” plaque [104,105]. The clear halo in the plaque’s center represents the phage’s lytic activity. The turbid ring surrounding the clear halo is formed by accumulating uninfected or partially infected host bacterial cells. These cells can resist phage infection (acquired resistance, defense systems) or have only been partially infected, potentially based on the aging of the bacterial lawn (non-infective after log phase), associated increases in the size of microcolonies making up the bacterial lawn, or because of less general phenomena such as the lysis inhibition phenotype [51,106]. However, it is important to note that phage plaque morphology can vary depending on the specific phage–host system and experimental conditions [107,108].

*Citrobacter* spp., classified as Gram-negative bacteria, are widely distributed in marine environments, including seawater, sediments, and marine eukaryotes [109,110,111,112]. Although little is known about the specific ecological roles of *Citrobacter* in marine environments, they are known to contribute to the degradation of organic matter [113]. *Staphylococcus* species in marine habitats play roles in various biological processes, such as biofilm formation and interactions with other marine microbes, and can indirectly contribute to nutrient cycling [114,115,116]. These bacteria often serve as indicators of pollution and pose public health risks, particularly due to their potential to harbor antibiotic resistance [117,118,119]. Frequently introduced into marine environments by human activities, both *Staphylococcus* and *Citrobacter* can persist and spread, impacting both ecosystem and human health [120,121]. As opportunistic pathogens, they can also cause infections in marine multicellular organisms [110,111,114,122,123]. However, *Staphylococcus* and *Citrobacter* species are generally not considered critical for the health of marine ecosystems [124,125]. Their presence in marine habitats is often linked to runoff or sewage discharge from human activities, rather than any significant ecological functions within the marine environment [126,127]. Consequently, virulent phages that infect these bacteria might help balance ecosystem and metaorganism homeostasis. Bacteriophages that target *Staphylococcus* and *Citrobacter* have been identified in marine environments and may play important roles in regulating bacterial populations [2,128]. These phages can control the abundance of their host bacteria, potentially limiting the spread of pathogenic or contaminant strains like *Staphylococcus aureus* and *Citrobacter* species [2,129]. By modulating the population sizes of these bacteria, phages contribute to microbial community dynamics, influence nutrient cycling, and help maintain ecological balance in marine ecosystems [129].

Moreover, the four isolated phages reflect narrow host range phages, infecting only a limited number of bacterial strains or species [80,130]. This specificity allows them to modulate bacterial populations by selectively limiting certain strains, such as *S. aureus*, which can otherwise spread uncontrollably and pose risks to marine ecosystems [131]. This targeted regulation is crucial for maintaining ecological balance and preventing the proliferation of potential pathogens [132]. Additionally, these phages can influence horizontal gene transfer by facilitating or inhibiting the movement of genetic material between bacteria, thereby affecting the spread of antibiotic resistance genes [133]. Due to these capabilities, narrow host range phages are valuable not only in natural ecosystems but also as potential tools in phage therapy to combat resistant infections [134,135]. Furthermore, their specificity makes them ideal candidates for bioengineering applications, such as designing targeted antimicrobials or developing biosensors [132,135,136]. 

All phages identified in this study showed effective and efficient lysis of *Staphylococcus* and *Citrobacter* by fast and effective binding of the phage to the host cells, short latency periods, and high burst sizes (Table 3 and Table 4). These characteristics are further important features for affecting natural microbiomes and are particularly relevant for potential therapeutic applications. Phage therapy uses intact natural phages or phage compounds to treat bacterial infections [137]. Due to the growing number of antibiotic-resistant bacterial species and the ban on the use of antibiotics in the aquatic environment [78,138,139,140], the interest in phage therapy particularly for aquaculture increased during the last few decades [141,142,143]. Phage therapy relies on extraordinary qualities of phages, including host specificity, self-replication, wide distribution, and safety [43,137,142,144]. Since phages are a natural way of managing bacterial infections, their usage does not contribute to the development of antibiotic resistance or the deposition of harmful residues in the environment. Finally, phages are versatile since they may be used alone or in cooperation with antibiotics or other therapies to improve their potency against bacterial infections. These features are entirely applicable in aquaculture, where traditional approaches to deal with pathogenic bacteria, such as antibiotics, are impossible [145,146]. Building on these advantages, the narrow host range of the isolated KMM1-KMM4 phages offers targeted therapeutic options in aquaculture, where specific bacterial infections require precise management [147,148]. These phages are effective against particular strains of *Staphylococcus* or *Citrobacter*, allowing for focused intervention without affecting non-target bacteria [129]. Their ability to selectively infect and reduce populations of specific pathogens minimizes the risk of disrupting beneficial microbial communities [136,149]. However, for these phages to be practical in aquaculture, their effectiveness must be verified against further various bacterial strains, and their stability, safety, and cost-effective production under various environmental conditions, such as pH and temperature, need to be thoroughly evaluated.

Lastly, our study demonstrated that phage research methods are still in their infancy, although several benchmarking studies on genomic and seasonal variation and diversity of tailed phages in the Baltic Sea were already published [84,85]. Bioinformatics tools specifically developed explicitly for phages still lag behind similar tools used in bacterial research [150,151,152]. However, some improved tools in recent years include Phaster, a web-based tool for identifying and annotating phage sequences in bacterial genomes and predicting their completeness [153]. VirSorter, VirFinder, and Phage AI implement machine learning to identify viral sequences in metagenomic datasets, distinguish between viral genomes, plasmids, and transposons, and predict phage host ranges and other characteristics [154,155]. In the present study, we used VICTOR for pairwise distance-based comparisons of the whole genome, vConTACT2 to determine the genus-level taxonomy of viral genomes, and VIRIDIC for calculating the intergenomic distance of viruses. The results obtained on species assignment of KMM2 and KMM4 using VICTOR and vConTACT2 demonstrate that phage softwares result in contrasting statements and must be improved. Using VICTOR and vConTACT2 did not allow for the differentiation of highly similar phages. Experimental data on differences within the lytic cycle and host range (Figure 2 and Figure 3), in combination with genome annotation, pointed to different species assignments of KMM2 and KMM4 (Figure 5). However, by improving estimates of phage genome similarity, particularly for distantly related phages, analyzing datasets including thousands of phage genomes, and creating an informative heatmap that includes not only the similarity values but also information about the genome lengths and aligned genome fraction, VIRIDIC finally confirmed taxonomic rank assignments (Appendix A and Appendix A). 

In summary, the present study describes the identification and characterization of four novel bacteriophages. Phages KMM2–4 infect *Citrobacter* and close relatives of *Enterobacteriaceae*, whereas phage KMM1 infected representatives of *Staphylococcus*, thus possessing a narrow host range. Although all identified phages demonstrated effective and efficient virulent properties relevant for phage application, future studies on the impact of those phages on the native microbiome of the moon jellyfish *A. aurita*, a model in metaorganism research, are of particular interest. Such studies may provide insights into the complex interdependence of phages and their bacterial hosts and how these relationships affect microbiomes to investigate the impact on the eukaryotic host *A. aurita*. It is conceivable that the infection of *A. aurita* with phages KMM1–4 might cause substantial changes in the bacterial community, potentially disrupting the multicellular host’s homeostasis. Even assuming that the phages impact *A. aurita*’s microbiome structure, it may be that the eukaryotic host has mechanisms to maintain its homeostasis. Future studies will focus on the characterization of attachment and infection mechanisms, and the impact of the identified phages on the microbiome and, consequently, the health of *A. aurita*.

## Figures and Tables

**Figure 1 viruses-15-01525-f001:**
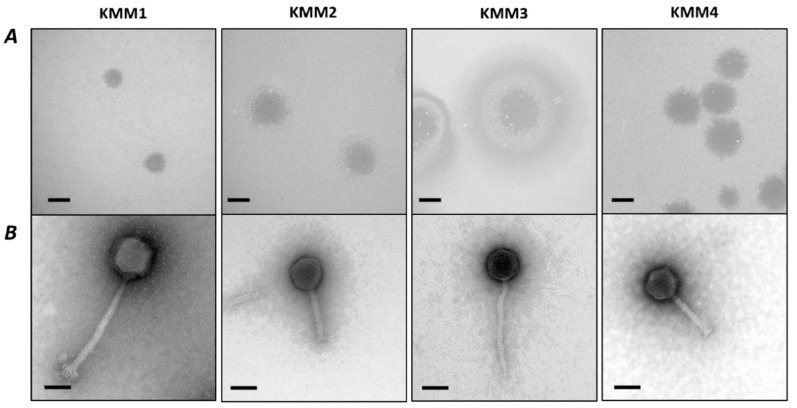
Plaque and virion morphology of isolated bacteriophages KMM1-KMM4. (**A**) Plaque morphologies were detected on MB double-agar layer plates after 16 h of incubation at 30 °C. Plaques formed on a lawn of *Staphylococcus* sp., PQ151711 (KMM1), *Citrobacter freundii,* OQ398153 (KMM2), and *Citrobacter* sp., OQ398154 (KMM3, KMM4). Scale bars represent 1 mm. (**B**) Transmission electron micrographs of phage lysates KMM1–KMM4, scale bars represent 50 nm.

**Figure 2 viruses-15-01525-f002:**
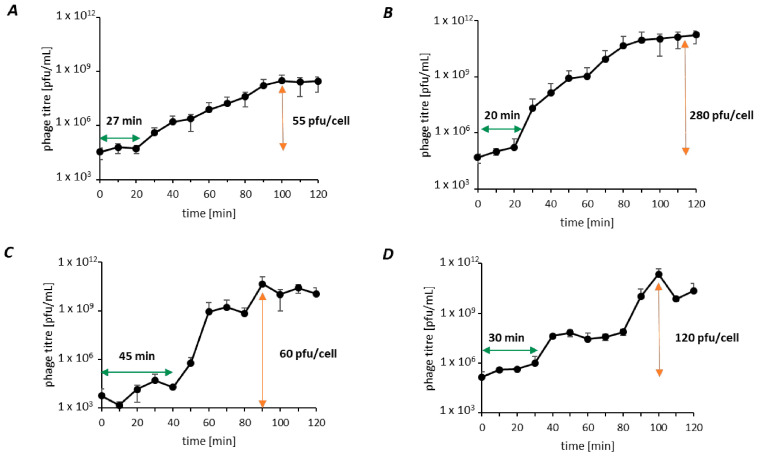
Infection cycles of isolated phages. One-step growth curves over 120 min were performed to calculate the latent period (green arrow) and burst size (orange arrow). (**A**) KMM1-infected *Staphylococcus* sp. (PQ151711) after 27 min with the release of 55 pfu/cell, (**B**) KMM2-infected *Citrobacter freundii* (OQ398153) after 20 min with the release of 280 pfu/cell, (**C**) KMM3- and (**D**) KMM4-infected *Citrobacter* sp. (OQ398154) after 45 and 30 min, respectively, with the release of 60 and 120 pfu/cell, respectively. Values represent the mean of three biological replicates.

**Figure 3 viruses-15-01525-f003:**
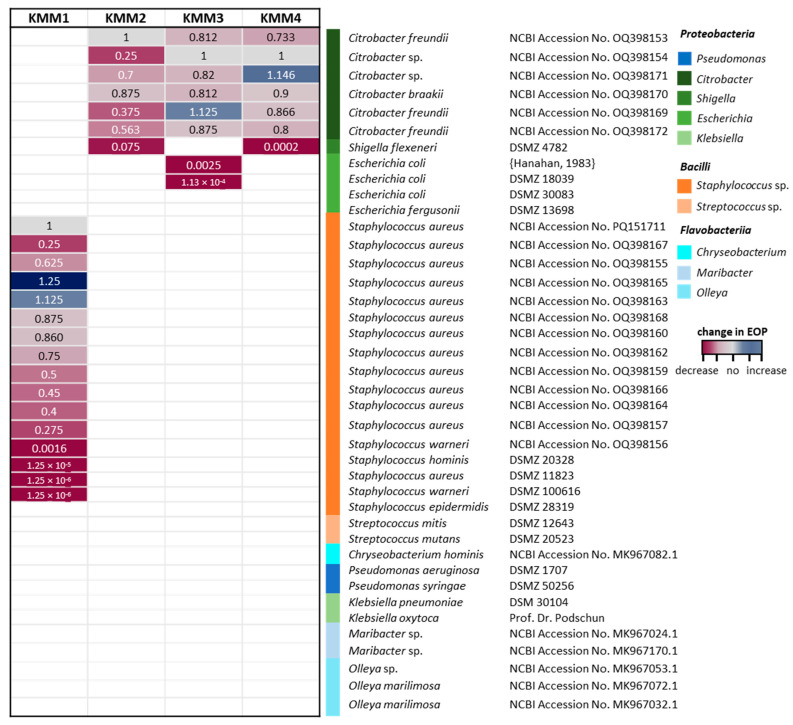
Host range of isolated phages. Phages KMM1–KMM4 were used for infection assays with selected taxons (color code on the right categorizes taxons into classes). The efficiency of plating (EOP) for each host bacterium was calculated by comparing it with a score of 10^9^ pfu/mL for the original host infection (value = 1). Missing coloring indicates no infection.

**Figure 4 viruses-15-01525-f004:**
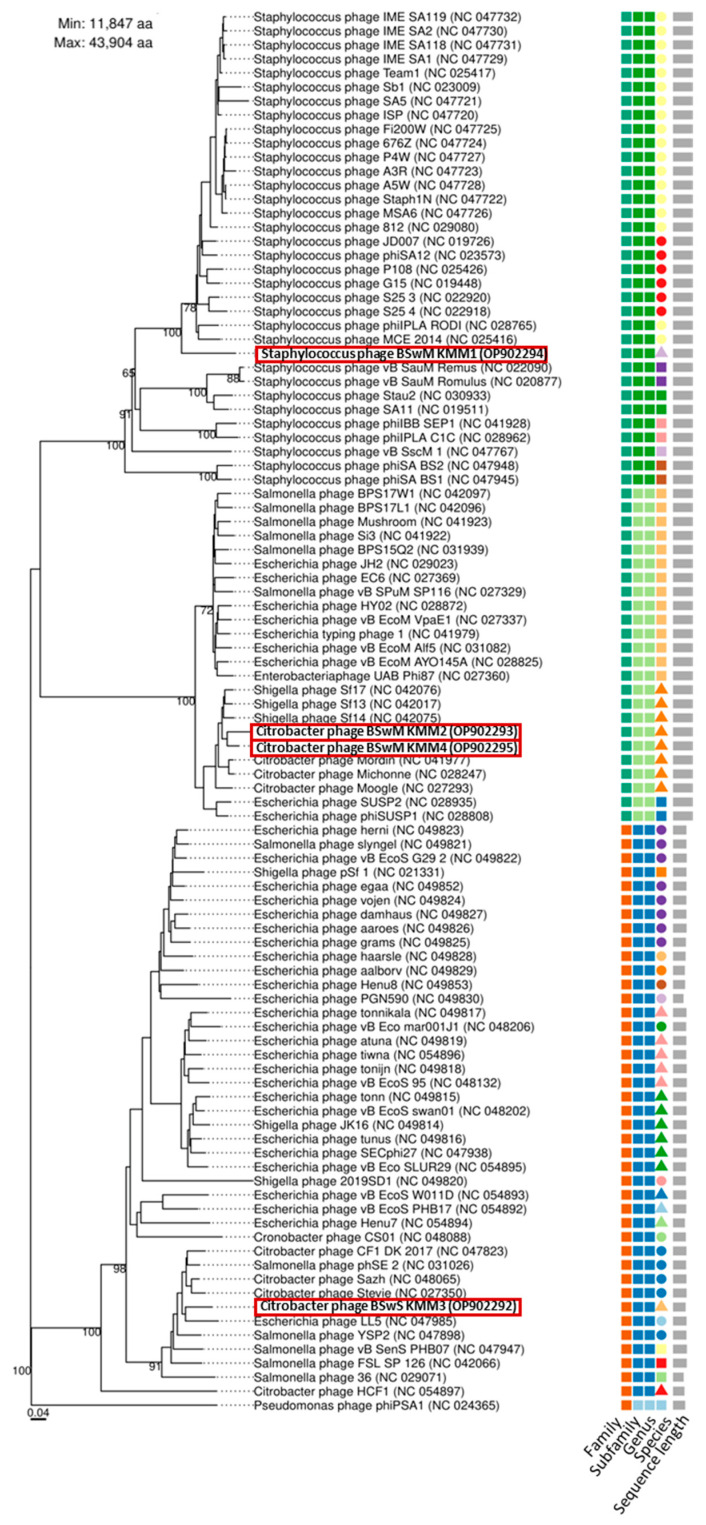
Taxonomic classification of phages KMM1–KMM4. Phylogenetic tree of isolated phages KMM1–KMM4 (red rectangles) was generated with the whole genome-based VICTOR analysis. Phages belonging to different families, subfamilies, genera, and species were color coded. The scale represents homology in %.

**Figure 5 viruses-15-01525-f005:**
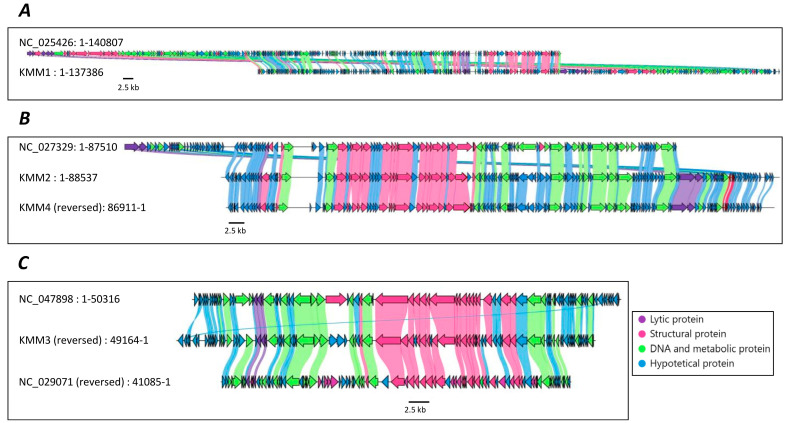
Genome annotations. The position and annotation of predicted viral genes in the phage genomes were visualized using Clinker. Coding domain sequences (CDS) are shown as arrows given the transcription direction; colors indicate their predicted function and amino acid sequence homologies to best homologs (nucleotide identity > 70%) are presented by corresponding alignments. (**A**) KMM1 (OP902294), (**B**) KMM2 (OP902295) and KMM4 (OP902293), and (**C**) KMM3 (OP902292).

**Table 1 viruses-15-01525-t001:** Bacterial strains used in this study. Bacterial strains were isolated in the study [22]. Isolates are sorted by the last column and phylum level. The column “Use in This study” refers to the use of the strains in the study. If not stated differently, the listed numbers in column “Reference” reflect NCBI Accession Numbers.

Strain No.	Strain	Reference	Phylum	Class	Order	Family	Source	Growth Medium	Growth Temp.	Use in This Study
74	*Micrococcus luteus*	MK967048.1	Actinomycetota	Actinomycetia	Micrococcales	Micrococcaceae	*A. aurita* polyp Baltic Sea husbandry	Marine Bouillon	30 °C	enrichment/ first screening
75	*Arthrobacter* sp.	MK967049.1	Actinomycetota	Actinomycetia	Micrococcales	Micrococcaceae	*A. aurita* polyp Baltic Sea husbandry	Marine Bouillon	30 °C	enrichment/ first screening
83	*Gordonia terrae*	MK967057.1	Actinomycetota	Actinomycetia	Mycobacteriales	Gordoniaceae	*A. aurita* polyp Baltic Sea husbandry	Marine Bouillon	30 °C	enrichment/ first screening
15	*Sulfitobacter* sp.	MK967015.1	Pseudomonadota	Alphaproteobacteria	Rhodobacterales	Rhodobacteraceae	*A. aurita* medusa Baltic Sea	Marine Bouillon	30 °C	enrichment/ first screening
20	*Sulfitobacter pontiacus*	MK967020.1	Pseudomonadota	Alphaproteobacteria	Rhodobacterales	Rhodobacteraceae	*A. aurita* medusa Baltic Sea husbandry	Marine Bouillon	30 °C	enrichment/ first screening
23	*Sulfitobacter* sp.	MK967023.1	Pseudomonadota	Alphaproteobacteria	Rhodobacterales	Rhodobacteraceae	*A. aurita* medusa Baltic Sea husbandry	Marine Bouillon	30 °C	enrichment/ first screening
69	*Rhodobacter* sp.	MK967043.1	Pseudomonadota	Alphaproteobacteria	Rhodobacterales	Rhodobacteraceae	*M. leidyi* Baltic Sea husbandry	Marine Bouillon	30 °C	enrichment/ first screening
78	*Sulfitobacter* sp.	MK967052.1	Pseudomonadota	Alphaproteobacteria	Rhodobacterales	Rhodobacteraceae	*A. aurita* polyp Baltic Sea husbandry	Marine Bouillon	30 °C	enrichment/ first screening
86	*Ruegeria* sp.	MK967060.1	Pseudomonadota	Alphaproteobacteria	Rhodobacterales	Rhodobacteraceae	*A. aurita* polyp Baltic Sea husbandry	Marine Bouillon	30 °C	enrichment/ first screening
89	*Ruegeria* sp.	MK967063.1	Pseudomonadota	Alphaproteobacteria	Rhodobacterales	Rhodobacteraceae	*A. aurita* polyp Baltic Sea husbandry	Marine Bouillon	30 °C	enrichment/ first screening
100	*Sulfitobacter* sp.	MK967074.1	Pseudomonadota	Alphaproteobacteria	Rhodobacterales	Rhodobacteraceae	*A. aurita* polyp North Sea husbandry	Marine Bouillon	30 °C	enrichment/ first screening
117	*Ruegeria mobilis*	MK967091.1	Pseudomonadota	Alphaproteobacteria	Rhodobacterales	Rhodobacteraceae	*A. aurita* polyp North Atlantic husbandry	Marine Bouillon	30 °C	enrichment/ first screening
147	*Phaeobacter gallaeciensis*	MK967120.1	Pseudomonadota	Alphaproteobacteria	Rhodobacterales	Rhodobacteraceae	Artificial Seawater 18 PSU	Marine Bouillon	30 °C	enrichment/ first screening
188	*Sulfitobacter pseudonitzschiae*	MK967160.1	Pseudomonadota	Alphaproteobacteria	Rhodobacterales	Rhodobacteraceae	Artificial Seawater 30 PSU	Marine Bouillon	30 °C	enrichment/ first screening
13	*Bacillus cereus*	MK967013.1	Bacillota	Bacilli	Bacillales	Bacillaceae	*A. aurita* medusa Baltic Sea	Marine Bouillon	30 °C	enrichment/ first screening
16	*Bacillus* sp.	MK967016.1	Bacillota	Bacilli	Bacillales	Bacillaceae	*A. aurita* medusa Baltic Sea	Marine Bouillon	30 °C	enrichment/ first screening
17	*Bacillus cereus*	MK967017.1	Bacillota	Bacilli	Bacillales	Bacillaceae	*A. aurita* medusa Baltic Sea	Marine Bouillon	30 °C	enrichment/ first screening
19	*Bacillus* sp.	MK967019.1	Bacillota	Bacilli	Bacillales	Bacillaceae	*A. aurita* medusa Baltic Sea husbandry	Marine Bouillon	30 °C	enrichment/ first screening
76	*Bacillus weihenstephanensis*	MK967050.1	Bacillota	Bacilli	Bacillales	Bacillaceae	*A. aurita* polyp Baltic Sea husbandry	Marine Bouillon	30 °C	enrichment/ first screening
85	*Staphylococcus warneri*	MK967059.1	Bacillota	Bacilli	Bacillales	Staphylococcaceae	*A. aurita* polyp Baltic Sea husbandry	Marine Bouillon	30 °C	enrichment/ first screening
88	*Staphylococcus* sp.	MK967062.1	Bacillota	Bacilli	Bacillales	Staphylococcaceae	*A. aurita* polyp Baltic Sea husbandry	Marine Bouillon	30 °C	enrichment/ first screening
73	*Enterococcus casseliflavus*	MK967047.1	Bacillota	Bacilli	Lactobacillales	Enterococcaceae	*A. aurita* polyp Baltic Sea husbandry	Marine Bouillon	30 °C	enrichment/ first screening
24	*Maribacter* sp.	MK967024.1	Bacteroidota	Flavobacteriia	Flavobacteriales	Flavobacteriaceae	*A. aurita* medusa Baltic Sea husbandry	Marine Bouillon	30 °C	enrichment/ first screening
57	*Olleya marilimosa*	MK967032.1	Bacteroidota	Flavobacteriia	Flavobacteriales	Flavobacteriaceae	*M. leidyi* Baltic Sea	Marine Bouillon	30 °C	enrichment/ first screening
79	*Olleya* sp.	MK967053.1	Bacteroidota	Flavobacteriia	Flavobacteriales	Flavobacteriaceae	*A. aurita* polyp Baltic Sea husbandry	Marine Bouillon	30 °C	enrichment/ first screening
181	*Chryseobacterium* sp.	MK967154.1	Bacteroidota	Flavobacteriia	Flavobacteriales	Weeksellaceae	Artificial Seawater 30 PSU	Marine Bouillon	30 °C	enrichment/ first screening
257	*Chryseobacterium* sp.	MK967218.1	Bacteroidota	Flavobacteriia	Flavobacteriales	Weeksellaceae	*M. leidyi* Baltic Sea husbandry	Marine Bouillon	30 °C	enrichment/ first screening
22	*Pseudolateromonas* sp.	MK967022.1	Pseudomonadota	Gammaproteobacteria	Alteromonadales	Pseudoalteromonadaceae	*A. aurita* medusa Baltic Sea husbandry	Marine Bouillon	30 °C	enrichment/ first screening
91	*Pseudoalteromonas prydzensis*	MK967065.1	Pseudomonadota	Gammaproteobacteria	Alteromonadales	Pseudoalteromonadaceae	*A. aurita* polyp Baltic Sea husbandry	Marine Bouillon	30 °C	enrichment/ first screening
101	*Pseudoalteromonas issachenkonii*	MK967075.1	Pseudomonadota	Gammaproteobacteria	Alteromonadales	Pseudoalteromonadaceae	*A. aurita* polyp North Sea husbandry	Marine Bouillon	30 °C	enrichment/ first screening
167	*Pseudoalteromonas* sp.	MK967140.1	Pseudomonadota	Gammaproteobacteria	Alteromonadales	Pseudoalteromonadaceae	Artificial Seawater 18 PSU	Marine Bouillon	30 °C	enrichment/ first screening
203	*Pseudoalteromonas espejiana*	MK967174.1	Pseudomonadota	Gammaproteobacteria	Alteromonadales	Pseudoalteromonadaceae	Artificial Seawater 30 PSU	Marine Bouillon	30 °C	enrichment/ first screening
219	*Pseudoalteromonas tunicata*	MK967188.1	Pseudomonadota	Gammaproteobacteria	Alteromonadales	Pseudoalteromonadaceae	*M. leidyi* Baltic Sea	Marine Bouillon	30 °C	enrichment/ first screening
224	*Pseudoalteromonas lipolytica*	MK967191.1	Pseudomonadota	Gammaproteobacteria	Alteromonadales	Pseudoalteromonadaceae	*M. leidyi* Baltic Sea	Marine Bouillon	30 °C	enrichment/ first screening
105	*Shewanella basaltis*	MK967079.1	Pseudomonadota	Gammaproteobacteria	Alteromonadales	Shewanellaceae	*A. aurita* polyp North Sea husbandry	Marine Bouillon	30 °C	enrichment/ first screening
21	*Cobetia amphilecti*	MK967021.1	Pseudomonadota	Gammaproteobacteria	Oceanospirillales	Halomonadaceae	*A. aurita* medusa Baltic Sea husbandry	Marine Bouillon	30 °C	enrichment/ first screening
55	*Marinomonas hwangdonensis*	MK967030.1	Pseudomonadota	Gammaproteobacteria	Oceanospirillales	Oceanospirillaceae	*M. leidyi* Baltic Sea	Marine Bouillon	30 °C	enrichment/ first screening
222	*Marinomonas pontica*	MK967189.1	Pseudomonadota	Gammaproteobacteria	Oceanospirillales	Oceanospirillaceae	*M. leidyi* Baltic Sea	Marine Bouillon	30 °C	enrichment/ first screening
262	*Oceanospirillaceae bacterium*	MK967222.1	Pseudomonadota	Gammaproteobacteria	Oceanospirillales	Oceanospirillaceae	*M. leidyi* Baltic Sea	Marine Bouillon	30 °C	enrichment/ first screening
11	*Pseudomonas* sp.	MK967012.1	Pseudomonadota	Gammaproteobacteria	Pseudomonadales	Pseudomonadaceae	*A. aurita* medusa Baltic Sea	Marine Bouillon	30 °C	enrichment/ first screening
90	*Pseudomonas putida*	MK967064.1	Pseudomonadota	Gammaproteobacteria	Pseudomonadales	Pseudomonadaceae	*A. aurita* polyp Baltic Sea husbandry	Marine Bouillon	30 °C	enrichment/ first screening
92	*Pseudomonas putida*	MK967066.1	Pseudomonadota	Gammaproteobacteria	Pseudomonadales	Pseudomonadaceae	*A. aurita* polyp Baltic Sea husbandry	Marine Bouillon	30 °C	enrichment/ first screening
93	*Pseudomonas* sp.	MK967067.1	Pseudomonadota	Gammaproteobacteria	Pseudomonadales	Pseudomonadaceae	*A. aurita* polyp Baltic Sea husbandry	Marine Bouillon	30 °C	enrichment/ first screening
94	*Pseudomonas* sp.	MK967068.1	Pseudomonadota	Gammaproteobacteria	Pseudomonadales	Pseudomonadaceae	*A. aurita* polyp Baltic Sea husbandry	Marine Bouillon	30 °C	enrichment/ first screening
132	*Pseudomonas fluorescens*	MK967106.1	Pseudomonadota	Gammaproteobacteria	Pseudomonadales	Pseudomonadaceae	Artificial Seawater 18 PSU	Marine Bouillon	30 °C	enrichment/ first screening
196	*Pseudomonas syringae*	MK967168.1	Pseudomonadota	Gammaproteobacteria	Pseudomonadales	Pseudomonadaceae	Artificial Seawater 30 PSU	Marine Bouillon	30 °C	enrichment/ first screening
77	*Vibrio anguillarum*	MK967051.1	Pseudomonadota	Gammaproteobacteria	Vibrionales	Vibrionaceae	*A. aurita* polyp Baltic Sea husbandry	Marine Bouillon	30 °C	enrichment/ first screening
80	*Vibrio anguillarum*	MK967054.1	Pseudomonadota	Gammaproteobacteria	Vibrionales	Vibrionaceae	*A. aurita* polyp Baltic Sea husbandry	Marine Bouillon	30 °C	enrichment/ first screening
18	*Staphylococcus aureus*	OQ398157	Bacillota	Bacilli	Bacillales	Staphylococcaceae	*A. aurita* medusa Baltic Sea husbandry	Marine Bouillon	30 °C	enrichment/ first screening
134	*Staphylococcus aureus*	OQ398164	Bacillota	Bacilli	Bacillales	Staphylococcaceae	Artificial Seawater 18 PSU	Marine Bouillon	30 °C	enrichment/ first screening
87	*Staphylococcus aureus*	OQ398160	Bacillota	Bacilli	Bacillales	Staphylococcaceae	*A. aurita* polyp Baltic Sea husbandry	Marine Bouillon	30 °C	enrichment/ first screening
14	*Staphylococcus warneri*	OQ398156	Bacillota	Bacilli	Bacillales	Staphylococcaceae	*A. aurita* medusa Baltic Sea	Marine Bouillon	30 °C	enrichment/ first screening
6	*Citrobacter freundii*	OQ398153	Pseudomonadota	Gammaproteobacteria	Enterobacterales	Enterobacteriaceae	*A. aurita* medusa Baltic Sea	Marine Bouillon	30 °C	enrichment/ first screening/ host range
7	*Citrobacter* sp.	OQ398154	Pseudomonadota	Gammaproteobacteria	Enterobacterales	Enterobacteriaceae	*A. aurita* medusa Baltic Sea	Marine Bouillon	30 °C	enrichment/ first screening/ host range
8	*Staphylococcus aureus*	PQ151711	Bacillota	Bacilli	Bacillales	Staphylococcaceae	*A. aurita* medusa Baltic Sea	Marine Bouillon	30 °C	enrichment/ first screening/ host range
62	*Sulfitobacter pontiacus*	OQ398158	Pseudomonadota	Alphaproteobacteria	Rhodobacterales	Rhodobacteraceae	*M. leidyi* Baltic Sea	Marine Bouillon	30 °C	host range
97	*Shewanella* sp.	OQ398161	Pseudomonadota	Gammaproteobacteria	Alteromonadales	Shewanellaceae	*A. aurita* polyp North Sea husbandry	Marine Bouillon	30 °C	host range
199	*Staphylococcus aureus*	OQ398168	Bacillota	Bacilli	Bacillales	Staphylococcaceae	Artificial Seawater 30 PSU	Marine Bouillon	30 °C	host range
DSMZ 11823	*Staphylococcus aureus*	DSMZ 11823	Bacillota	Bacilli	Bacillales	Staphylococcaceae	clinical material	Trypticase Soy Yeast Broth	37 °C	host range
67	*Staphylococcus aureus*	OQ398159	Bacillota	Bacilli	Bacillales	Staphylococcaceae	*M. leidyi* Baltic Sea husbandry	Marine Bouillon	30 °C	host range
102	*Staphylococcus aureus*	OQ398162	Bacillota	Bacilli	Bacillales	Staphylococcaceae	*A. aurita* polyp North Sea husbandry	Marine Bouillon	30 °C	host range
158	*Staphylococcus aureus*	OQ398165	Bacillota	Bacilli	Bacillales	Staphylococcaceae	Artificial Seawater 18 PSU	Marine Bouillon	30 °C	host range
161	*Staphylococcus aureus*	OQ398166	Bacillota	Bacilli	Bacillales	Staphylococcaceae	Artificial Seawater 18 PSU	Marine Bouillon	30 °C	host range
127	*Staphylococcus aureus*	OQ398163	Bacillota	Bacilli	Bacillales	Staphylococcaceae	*A. aurita* polyp North Atlantic husbandry	Marine Bouillon	30 °C	host range
DSMZ 28319	*Staphylococcus epidermidis*	DSMZ 28319	Bacillota	Bacilli	Bacillales	Staphylococcaceae	catheter sepsis	Trypticase Soy Yeast Broth	37 °C	host range
DSMZ 20328	*Staphylococcus hominis*	DSMZ 20328	Bacillota	Bacilli	Bacillales	Staphylococcaceae	human skin	Trypticase Soy Yeast Broth	37 °C	host range
DSMZ 100616	*Staphylococcus warneri*	DSMZ 100616	Bacillota	Bacilli	Bacillales	Staphylococcaceae	cleanroom facility, TAS	Trypticase Soy Yeast Broth	30 °C	host range
DSMZ 12643	*Streptococcus mitis*	DSMZ 12643	Bacillota	Bacilli	Lactobacillales	Streptococcaceae	oral cavity, human	Trypticase Soy Yeast Broth	37 °C	host range
DSMZ 20523	*Streptococcus mutans*	DSMZ 20523	Bacillota	Bacilli	Lactobacillales	Streptococcaceae	carious dentine	Trypticase Soy Yeast Broth	37 °C	host range
296	*Citrobacter braakii*	OQ398170	Pseudomonadota	Gammaproteobacteria	Enterobacterales	Enterobacteriaceae	*A. aurita* polyp North Atlantic husbandry	Marine Bouillon	30 °C	host range
283	*Citrobacter freundii*	OQ398169	Pseudomonadota	Gammaproteobacteria	Enterobacterales	Enterobacteriaceae	*A. aurita* polyp Baltic Sea husbandry	Marine Bouillon	30 °C	host range
321	*Citrobacter freundii*	OQ398172	Pseudomonadota	Gammaproteobacteria	Enterobacterales	Enterobacteriaceae	Artifical Seawater 30 PSU	Marine Bouillon	30 °C	host range
313	*Citrobacter* sp.	OQ398171	Pseudomonadota	Gammaproteobacteria	Enterobacterales	Enterobacteriaceae	Artifical Seawater 18 PSU	Marine Bouillon	30 °C	host range
DSMZ 18039	*Escherichia coli*	DSMZ 18039	Pseudomonadota	Gammaproteobacteria	Enterobacterales	Enterobacteriaceae	unknown source	Luria-Bertani Bouillon	37 °C	host range
strain 8	*Escherichia coli*	[47]	Pseudomonadota	Gammaproteobacteria	Enterobacterales	Enterobacteriaceae	unknown source	Luria-Bertani Bouillon	37 °C	host range
DSMZ 30083	*Escherichia coli*	DSMZ 30083	Pseudomonadota	Gammaproteobacteria	Enterobacterales	Enterobacteriaceae	urine	Luria-Bertani Bouillon	37 °C	host range
DSMZ 13698	*Escherichia fergusonii*	DSMZ 13698	Pseudomonadota	Gammaproteobacteria	Enterobacterales	Enterobacteriaceae	faeces of 1-year-old boy	Luria-Bertani Bouillon	37 °C	host range
strain 27	*Klebsiella oxytoca*	Prof. Dr. Podschun, (National Reference Laboratory for Klebsiella species, Kiel University)	Pseudomonadota	Gammaproteobacteria	Enterobacterales	Enterobacteriaceae	unknown source	Nutrient Broth	30 °C	host range
DSMZ 30104	*Klebsiella pneumoniae*	DSMZ 30104	Pseudomonadota	Gammaproteobacteria	Enterobacterales	Enterobacteriaceae	unknown source	Nutrient Broth	30 °C	host range
DSMZ 4782	*Shigella flexeneri*	DSMZ 4782	Pseudomonadota	Gammaproteobacteria	Enterobacterales	Enterobacteriaceae	unknown source	Caso Bouillon	37 °C	host range
DSMZ 1707	*Pseudomonas aeruginosa*	DSMZ 1707	Pseudomonadota	Gammaproteobacteria	Pseudomonadales	Pseudomonadaceae	unknown source	Caso Bouillon	30 °C	host range
9	*Staphylococcus aureus*	OQ398155	Bacillota	Bacilli	Bacillales	Staphylococcaceae	*A. aurita* medusa Baltic Sea	Marine Bouillon	30 °C	host range
170	*Staphylococcus aureus*	OQ398167	Bacillota	Bacilli	Bacillales	Staphylococcaceae	Artificial Seawater 18 PSU	Marine Bouillon	30 °C	host range
DSMZ 50256	*Pseudomonas syringae*	DSMZ 50256	Pseudomonadota	Gammaproteobacteria	Pseudomonadales	Pseudomonadaceae	*Triticum aestivum*, glume rot of wheat	Caso Bouillon	30 °C	host range
24	*Maribacter* sp.	MK967024.1	Bacteroidota	Flavobacteriia	Flavobacteriales	Flavobacteriaceae	*A. aurita* medusa Baltic Sea husbandry	Marine Bouillon	30 °C	host range
79	*Olleya* sp.	MK967053.1	Bacteroidota	Flavobacteriia	Flavobacteriales	Flavobacteriaceae	*A. aurita* polyp Baltic Sea husbandry	Marine Bouillon	30 °C	host range
98	*Olleya marilimosa*	MK967072.1	Bacteroidota	Flavobacteriia	Flavobacteriales	Flavobacteriaceae	*A. aurita* polyp North Sea husbandry	Marine Bouillon	30 °C	host range
108	*Chryseobacterium hominis*	MK967082.1	Bacteroidota	Flavobacteriia	Flavobacteriales	Flavobacteriaceae	*A. aurita* polyp North Sea husbandry	Marine Bouillon	30 °C	host range
57	*Olleya marilimosa*	MK967032.1	Bacteroidota	Flavobacteriia	Flavobacteriales	Flavobacteriaceae	*M. leidyi* Baltic Sea	Marine Bouillon	30 °C	host range
199	*Maribacter* sp.	MK967170.1	Bacteroidota	Flavobacteriia	Flavobacteriales	Flavobacteriaceae	Artificial Seawater 30 PSU	Marine Bouillon	30 °C	host range

**Table 2 viruses-15-01525-t002:** Viral genome characteristics and overview of assembly-related metrics.

Phage	NCBI Accession No.	No. of Reads	No. of Filtered Reads	Sequence Coverage	N50	Genome Length (bps)	GC Content (%)	Predicted ORFs	Unknown Proteins
***Staphylococcus* phage BSwM KMM1**	OP902294	4.085	3.214	247.488	17.553	137.386	31.77	259	200
***Citrobacter* phage BSwM KMM2**	OP902295	810	595	74.163	22.118	88.537	39.55	137	94
***Citrobacter* phage BSwS KMM3**	OP902292	837	598	130.676	20.517	49.164	43.17	92	58
***Citrobacter* phage BSwM KMM4**	OP902293	6.433	5.371	544.327	23.894	86.911	39.02	138	100

**Table 3 viruses-15-01525-t003:** Characteristics of isolated bacteriophages KMM1–KMM4. Plaque properties (N = 10), phage morphology (tail, N= 20; head, N = 10), latent period, and burst size (N = 3) are listed.

	KMM1	KMM2	KMM3	KMM4
Plaques properties	Small, round, and clear Size: 0.8–1 mm	Small, round, and clear Size: 1–1.5 mm	Big, round with a clear center and surrounding turbid halo Size: 2.5–3.5 mm	Small, round, and clear Size: 1.2–1.5 mm
Phage morphology Tail length (nm) Head width (nm) Head shape	176.4 ± 8.4 75.8 ± 3 icosahedral	90.9 ± 3.9 58.2 ± 2.7 icosahedral	131.7 ± 9.3 50 ± 1.9 icosahedral	90.5 ± 4.5 57.5 ± 2.8 icosahedral
Latent period (min)	27	20	45	30
Burst size (phages/infected bacterial cell)	55	280	60	120
Taxonomy prediction	Caudoviricetes (*Myovirus*-like)	Caudoviricetes (*Myovirus*-like)	Caudoviricetes (*Siphovirus*-like)	Caudoviricetes (*Myovirus*-like)

**Table 4 viruses-15-01525-t004:** Adsorption dynamics of bacteriophages KMM1–KMM4. Adsorption rates were determined 5 min after phage addition to the primary hosts (*Staphylococcus* sp., *Citrobacter freundii*, and *Citrobacter* sp.). The number of phages adsorbed to the cells generated a decrease in phage titer. The percentage of adsorbed phages and the adsorption constant (*k*) were calculated. Values are the mean of three biological replicates with corresponding standard deviations.

Bacteriophage	Phage Titre, 0 min (pfu/mL)	Phage Titre, 5 min (pfu/mL)	% of Adsorbed Phages	Adsorption Constant, *k* (mL/min)
KMM1	2.3 × 10^7^ ± 1.2 × 10^7^	3.4 × 10^4^ ± 3.9 × 10^4^	99.9	1.08 × 10^−7^
KMM2	3.1 × 10^7^ ± 2.0 × 10^7^	4.8 × 10^4^ ± 1.2 × 10^4^	99.9	4.62 × 10^−8^
KMM3	2.0 × 10^7^ ± 1.3 × 10^7^	5.7 × 10^3^ ± 4.7 × 10^2^	99.8	1.26 × 10^−7^
KMM4	4.5 × 10^7^ ± 2.8 × 10^7^	1.4 × 10^5^ ± 1.3 × 10^5^	99.8	8.70 × 10^−8^

## Data Availability

The phage genomes are available at NCBI under the accession numbers OP902294: *Staphylococcus* phage BSwM KMM1, OP902295: *Citrobacter* phage BSwM KMM2, OP902292: *Citrobacter* phage BSwS KMM3, and OP902293: *Citrobacter* phage BSwM KMM4. Sequences were deposited on the Sequence Read Archive (SRA) under BioProject PRJNA908753 and raw sequence accession numbers SRR22580853, SRR22580850, SRR22580849, and SRR22580845. Bacterial sequence data derived from Sanger sequencing of the 16S rRNA genes of bacterial isolates is deposited under GenBank accession numbers OQ397638–OQ397653, OQ398153–OQ398172, and PQ151711.

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
