# Peer review of "Four Novel Caudoviricetes Bacteriophages Isolated from Baltic Sea Water Infect Colonizers of Aurelia aurita"

_viruses, 2023, doi:10.3390/v15071525_

Round 1

Reviewer 1 Report

In this study, starting from sea water, the authors isolated 4 different bacteriophages infecting bacteria that are part of the microbiota from the moon jellyfish Aurelia aurita. This jellyfish is highly studied and it represents an important biological model. It has indeed a wide distribution across oceans and it is considered as a key player in marine ecosystems. Among the 4 described bacteriophages, one has a broad host range, infecting both gram-positive (Staphylococcus) and gram-negative hosts (Pseudomonas). The other three present a narrower host range (only gram-negative, mostly, but not exclusively, Citrobacter strains). The host range has been evaluated on a collection of 43 strains, previously isolated by the authors (Weiland-Bräuer et al., 2020) from A. aurita microbiota.

Overall, the study is clearly presented and well conducted. The isolation of these 4 phages, with contrasted host ranges, will be of high interest for further studies on the role of phages on metaorganisms. The authors moreover mention perspectives for phage therapy in aquaculture.

Main comments

1/ In the introduction and throughout the manuscript, I suggest to replace “lytic” by “virulent” and “lysogenic” by “temperate”, whenever relevant. Indeed, virulent phages undergo only lytic cycles whereas temperate phages undergo either lytic of lysogenic cycles, according to the circumstances. It is therefore more adequate to use “temperate phage” and “virulent phage”, rather than “lytic phage” and “lysogenic phage”.

2/ In the introduction and throughout the manuscript, I suggest to replace “Myovirus” and “Siphovirus” by “myovirus-like” and “siphovirus like”. At the first occurrence, it could be worth mentioning that “myovirus-like” viruses have a long contractile tail, and that “siphovirus-like viruses” have a long, non-contractile tail.

Indeed, to the best of my knowledge, Myovirus and Siphovirus have been suppressed from the official taxonomy. In particular, in the abstract, line 19, “phage families Siphovirus (KMM3) and Myovirus (KMM1/2/4)” is not correct, since Siphovirus and Myovirus are not viral families.

3/ Most of the important parameters of phage life history were experimentally determined in the study: lifestyle, host range and efficiency of plating, latent period, as well as burst size. The adsorption rate was not measured.

Although the experimental work is overall very robust, I have a concern regarding the determination of the burst size. It was measured at a multiplicity of infection (moi) of 1, whereas a low moi (typically 0.01 to 0.1) is recommended (see for instance: https://www.protocols.io/view/One-step-growth-experiments-bacteriophages-x54v9mzpg3eq/v1). Indeed, although the average number of phage particle per cell if of 1 (at a moi of 1), there is some variability, so that some cells will not be infected at all, whereas some others will be infected by more than one phage particles. The total estimate of infected cells becomes less than the phage input. Moreover, when cells are infected by more than one phages, the infectious cycle may be affected, creating an additional bias. I wonder whether the choice of this relatively “high” moi may explain the appearance of some growth curves obtained by the authors, showing a not so clear pattern.

I suggest to explain/discuss this limitation in the text of the manuscript, in the result section.

4/ The bioinformatic analysis has been conducted with state-of-the-art tools, such as VIRIDIC, vConTACT2and VICTOR. Their outcome is adequately analyzed and discussed, in particular regarding the possibility to identify viral species.

5/ In the Abstract (line 16) and elsewhere in the manuscript, the burst size is expressed in PFU/mL. It would be more meaningful to express it in PFU/cell.

Minor comments

- line 386: replace “109” by “109

- line 535, replace “genera” by “types” (Gram minus and Gram plus) to avoid any confusion with the notion of taxonomic genus (?)

- line 537 : replace “KMM1 might has evolved” by “KMM1 might have evolved”

Reviewer 2 Report

Peer Review for Viruses

Stante et al

Title

Four novel Caudoviricetes bacteriophages isolated from Baltic Sea water infect colonizers of Aurelia aurita.

Summary

Here, the authors report the identification and characterization of four novel bacteriophages (BsWM KMM1–4). The authors frame the biological importance of these phages as a part of the Aurelia aurita-centered metaorganism. KMM1 infected both Gram-negative (Pseudomonas) and Gram-positive (Coagulase positive Staphylococcus) hosts with equivalent efficiency, and retaining the coagulase negative disparity observed for other Staphylococcal phages.

Comments

Despite the promising introduction focused on the metaorganism, the results were limited to the wrote formula of SEAphage isolation and description. Had the phages been isolated from the metaorganism there would be a direct connection, but these came from shallow seawater rendering the association with the metaorganism circumstantial at best. In a classic description of the earthworm metaorganism, Vince Fischetti and colleagues showed a lysogenic phage switched its B. anthracis host to a soil-dwelling lifestyle.

Since lytic phages affect their metaorganism through bacterial turnover, is there a brief experiment the authors could add to test the impact of each of their phages on the health or productivity of the metaorganism? Alternatively, could an experiment be added to describe if each phage causes the host bacterial population to undulate or conversely draw a bloom down to a low but steady state?

KMM3 was noted as forming large clear plaques with a turbid halo, but no analysis or interpretation followed. Please add this to the discussion or consider a brief experiment on this phage that describes its impact on the metaorganism.

Overall the manuscript is well written and just needs a firm tie-in to the metaorganism.
